# Machine learning-guided co-optimization of fitness and diversity facilitates combinatorial library design in enzyme engineering

Kerr Ding [1,5], Michael Chin [2,5], Yunlong Zhao [2,5], Wei Huang[2], Binh Khanh Mai [3], Huanan Wang[2], Peng Liu [3] ✉, Yang Yang [2,4] ✉ & Yunan Luo [1] ✉

The effective design of combinatorial libraries to balance fitness and diversity facilitates the engineering of useful enzyme functions, particularly those that are poorly characterized or unknown in biology. We introduce MODIFY, a machine learning (ML) algorithm that learns from natural protein sequences to infer evolutionarily plausible mutations and predict enzyme fitness. MODIFY co-optimizes predicted fitness and sequence diversity of starting libraries, prioritizing high-fitness variants while ensuring broad sequence coverage. In silico evaluation shows that MODIFY outperforms state-of-the-art unsupervised methods in zero-shot fitness prediction and enables ML-guided directed evolution with enhanced efficiency. Using MODIFY, we engineer generalist biocatalysts derived from a thermostable cytochrome *c* to achieve enantioselective C-B and C-Si bond formation via a new-to-nature carbene transfer mechanism, leading to biocatalysts six mutations away from previously developed enzymes while exhibiting superior or comparable activities. These results demonstrate MODIFY's potential in solving challenging enzyme engineering problems beyond the reach of classic directed evolution.

Biocatalysis has emerged as a promising alternative for the sustainable production of specialty chemicals and pharmaceuticals, due to the substantial reaction acceleration and exquisite stereocontrol imposed by enzyme catalysts under mild and environmentally friendly conditions[1–3]. Over the past several decades, the advent of directed evolution[4,5] has enabled the development of customized enzymes with excellent catalytic activity and stereoselectivity, which can often surpass those of small-molecule catalysts[6]. Despite the tremendous success of laboratory-directed evolution in enzyme engineering[4,5], its efficiency in navigating the vast fitness landscape remains limited. In recent years, machine learning (ML) methods have emerged as a powerful strategy for accelerating enzyme engineering[7–9]. Supervised ML models have been trained to learn the relationships between

protein sequences and properties[10–13]. Acting as surrogates for laboratory screening, these ML models expedite enzyme engineering through in silico fitness prediction and prioritization of enzyme variants, thus reducing the overall experimental burden[10,14–16].

Herein, we tackle a more challenging problem in enzyme engineering: advancing ML algorithms for the development of synthetically valuable enzyme functions that are not known in biology. Over the past decade, drawing inspiration from synthetic chemistry, biocatalysis researchers have developed an array of enzymatic activities that are not encountered in the biological world[17–20], including atom transfer radical cyclase[21] and radical pyridoxal enzyme[22] activities developed by our team. However, general ML strategies to streamline the discovery of new-to-nature enzyme activities remain

[1]School of Computational Science and Engineering, Georgia Institute of Technology, Atlanta, GA 30332, USA. [2]Department of Chemistry and Biochemistry, University of California, Santa Barbara, CA 93106, USA. [3]Department of Chemistry, University of Pittsburgh, Pittsburgh, PA 15260, USA. [4]Biomolecular Science and Engineering (BMSE) Program, University of California, Santa Barbara, CA 93106, USA. [5]These authors contributed equally: Kerr Ding, Michael Chin, Yunlong Zhao. ✉e-mail: pengliu@pitt.edu; yang@chem.ucsb.edu; yunan@gatech.edu

underdeveloped. Due to the scarcity of fitness data for new-to-nature enzyme functions, training of supervised ML models to guide such directed evolution efforts is challenging. Thus, central to the successful engineering of new-to-nature enzymes is the design of effective starting libraries without relying on experimentally determined enzyme fitness, allowing for the identification of initial hits as well as downstream optimization of highly functional enzyme variants. While large-scale random sampling of combinatorial mutants has been used to generate diversity, the inevitable inclusion of deleterious mutations into random combinatorial variants results in large quantities of non-functional mutants and reduces the utility of such libraries. To address these limitations, an effective starting library should achieve two desiderata—high fitness and rich diversity. The former ensures the identification of excellent starting variants for further enzyme engineering, while the latter increases the likelihood of uncovering multiple fitness peaks. Moreover, due to the strength of supervised ML models in interpolation rather than extrapolation, a diverse starting library enriched with distinct functional variants will allow ML models to more efficiently map out the fitness landscape, ultimately enhancing the efficiency of downstream ML-guided directed evolution (MLDE). Nevertheless, designing an effective starting library with co-optimized fitness and diversity remains a challenging task.

In this work, we introduce MODIFY (ML-optimized library design with improved fitness and diversity), an ML algorithm for effective starting library design in enzyme engineering. Given as input a set of residues for enzyme engineering, MODIFY designs high-quality libraries to sample variants from the combinatorial sequence space that are more likely to be functional, while maintaining a high level of library diversity. To address the cold-start challenge where no experimentally characterized fitness data is available, MODIFY leverages pre-trained unsupervised models to develop an ensemble model for zero-shot fitness predictions. MODIFY libraries achieve a Pareto optimal balance between the expected fitness and the sequence diversity in that neither of the two metrics can be further improved without decreasing the other. Benchmarked on 87 protein deep mutational scanning (DMS) datasets, MODIFY achieves robust and accurate zero-shot fitness predictions, outperforming several state-of-the-art unsupervised ML methods. Further evaluated on the experimentally characterized fitness landscape of the GB1 protein, MODIFY provided a diverse combinatorial library enriched with high-fitness variants. In silico MLDE experiments show that MODIFY libraries more effectively map out the sequence space and delineate higher-fitness regions, offering a more informative training set for effective MLDE. Furthermore, we applied MODIFY to design effective libraries for the rapid generation of novel, efficient, and stereoselective new-to-nature biocatalysts for the stereoselective construction of C−B and C−Si bonds. Notably, the top-performing enzyme variants derived from the MODIFY-designed library are distinct from experimentally evolved ones, establishing fertile ground for the further understanding of enzyme structure-activity relationships. Moreover, generalist biocatalysts that catalyze both the C−B and the C−Si bond formation were identified from MODIFY library, further highlighting the utility of this MODIFY library design algorithm in new-to-nature enzyme engineering.

## Results

### Overview of MODIFY, an ML algorithm to co-optimize library fitness and diversity

We developed MODIFY (Fig. 1), an ML-guided framework to design high-fitness, high-diversity enzyme libraries for the engineering of new enzyme functions. Given a set of specified residues in a parent enzyme, MODIFY affords combinatorial mutant libraries to strike a balance between maximal expected fitness and diversity, without requiring functionally characterized mutants as prior knowledge (Fig. 1a). MODIFY applies a novel ensemble ML model that leverages protein language models (PLMs) and sequence density models to make zero-shot fitness predictions and employs a Pareto optimization scheme to

design libraries with both high expected fitness and high diversity. High levels of expected fitness ensure the effective sampling of functional enzyme variants, while the high diversity of designed enzyme libraries spanning a wide sequence space allows the exploration of new enzyme variants. Balancing fitness and diversity is achieved by solving the optimization problem: max fitness + $\lambda \cdot$ diversity, with parameter $\lambda$ balancing between prioritizing high-fitness variants (exploitation) and generating a more diverse sequence set (exploration). In this way, MODIFY traces out an optimal tradeoff curve known as the Pareto frontier, on which each point represents an optimal library where neither desiderata can be further improved without compromising the other (Fig. 1b). To refine the library, enzyme variants sampled from the library are further filtered based on protein foldability and stability (Fig. 1c). Applying MODIFY to design a high-quality starting library for both new-to-nature borylation and silylation, we identified a generalist biocatalyst with substantially altered loop dynamics from top-performing MODIFY variants (Fig. 1d).

### Accurate zero-shot fitness prediction

We first assessed the zero-shot fitness prediction ability of MODIFY using the ProteinGym benchmark dataset[23], which comprises 87 DMS assays that provide experimental measurements of protein fitness, spanning different functions such as enzyme catalytic activity, binding affinity, stability, and growth rate. This benchmark thus represents a holistic evaluation of MODIFY's zero-shot fitness prediction across various protein families and functions. We compared MODIFY with its constituent models: two state-of-the-art PLMs (ESM-1v[24] and ESM-2[25]), two MSA-based sequence density models (EVmutation[26] and EVE[27]), and a hybrid PLM that incorporated MSA data (MSA Transformer[28]). These individual models were previously established as effective unsupervised predictors in protein fitness and disease variant effect prediction[24,27]. We found that no single baseline consistently outperformed the others. In contrast, MODIFY's ensemble predictor stood out by delivering accurate and robust predictions (Fig. 2a) and achieving the best Spearman correlation for the largest number of times (34/87; Fig. 2b). Across ProteinGym, MODIFY consistently outperformed at least one of the baselines in all 87 DMS datasets and often ranked at or near the top (Fig. 2a). As ProteinGym covers a wide array of protein families, this result demonstrates the general utility of MODIFY in zero-shot fitness prediction across a wide range of proteins.

Additionally, stratifying results based on the MSA depth of parent sequences in ProteinGym indicated that MODIFY outperformed all baselines for proteins across low, medium, and high MSA depths (Fig. 2c; Supplementary Information A.1). In contrast, no single baseline consistently outperformed other baselines across the three categories. These results underscore MODIFY's capacity to provide reliable fitness predictions for diverse protein families, including those lacking ample homologous sequences, highlighting its general applicability. We further compared MODIFY with the baseline methods on the latest release (v1.0) of the ProteinGym benchmark dataset[29] with 217 DMS assays (Supplementary Information A.1). The results mirrored these findings (Supplementary Figs. 1 and 2), featuring the superior zero-shot protein fitness capability of MODIFY across a diverse array of proteins. It should also be noted that MODIFY achieved the highest zero-shot protein fitness prediction for DMS assays measuring catalytic or related biochemical activities (Supplementary Fig. 2b), further highlighting the suitability of MODIFY for enzyme engineering.

Since the majority of DMS datasets in ProteinGym focused on single mutants, we further examined MODIFY's fitness prediction ability for high-order mutants using the experimentally characterized fitness landscapes of three proteins, including GB1[30], ParD3[31,32], and CreiLOV[33], covering combinatorial mutation spaces of 4, 3, and 15 residues, respectively. MODIFY achieved notable performance improvements over other baselines, suggesting its generalizability in predicting the fitness of high-order mutants (Supplementary Fig. 3).

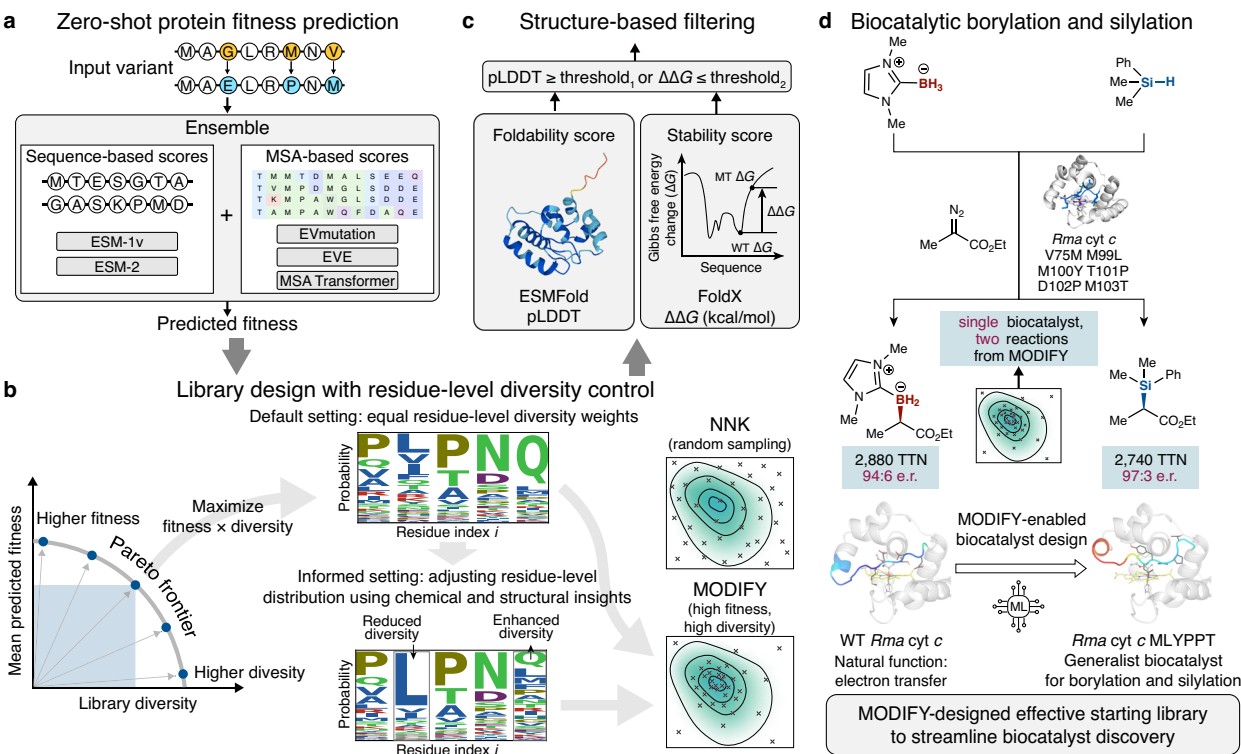

**Fig. 1 | MODIFY: an ML-guided framework for the design of enzyme engineering starting libraries with both high fitness and high diversity. a** MODIFY leverages pre-trained protein language models and multiple sequence alignment (MSA)-based sequence density models to build an ensemble ML model for zero-shot fitness predictions, effectively eliminating evolutionarily unfavorable variants. **b** MODIFY co-optimizes the library's diversity and predicted fitness, pinpointing the Pareto optimal balance between the two. MODIFY offers diversity control at a residue resolution, enabling researchers to either explore a diverse range of amino acids or focus on a subset of compatible amino acids based on biophysical and biochemical insights. **c** MODIFY further performs a quality control step to filter out problematic variants in the library based on protein foldability (ESMFold pLDDT) and stability (FoldX ΔΔG). **d** MODIFY-enabled discovery of effective generalist biocatalysts for enantioselective new-to-nature borylation and silylation.

Taken together, these results demonstrate the superior accuracy and robustness of MODIFY in predicting variant fitness across a diverse range of protein families, which lays the groundwork for the library design algorithm detailed below.

## In silico evaluation of starting library design on GB1

With these benchmark results in hand, we next applied MODIFY to optimize a starting library on a four-site combinatorial sequence space for the GB1 protein (Fig. 3a). The fitness landscape of these sites (V39, D40, G41, and V54) was previously mapped out experimentally[30], where the fitness was defined by both stability (fraction of folded proteins) and function (binding affinity to IgG-Fc). This experimentally derived dataset allowed for a retrospective assessment of the quality of our MODIFY library.

A unique strength of MODIFY is the optimization of the composition diversity of amino acids at the residue-level resolution, controlled by a diversity hyperparameter $\alpha_i$ for residue $i$ (Methods), which generalizes previous methods that only optimize diversity at the sequence level[34,35]. Here, we first applied MODIFY's default setting (denoted as MODIFY) to design the library, assigning equal diversity weights $\alpha_i$ to all four sites (Fig. 1b and Methods). MODIFY afforded a library striking an optimal balance between library diversity and mean predicted fitness of the library (Fig. 3b). By contrast, the commonly used NNK library produced a library with high diversity but low mean predicted fitness. Upon assessing a 500-sequence library designed by MODIFY and NNK using ground-truth fitness data, the MODIFY library exhibited higher mean experimental fitness (Fig. 3c) and preserved library diversity as indicated by average entropy (Fig. 3d). In contrast, the NNK library—although marginally more diverse—was predominantly populated with nonfunctional

variants (Fig. 3d), as indicated by its minimal mean experimental fitness, which was similar to that of a control library that samples sequences uniformly at random (Fig. 3c). Importantly, MODIFY's improvements were consistently observed across varying library sizes (Fig. 3e). Additionally, Exploitation, a variation of MODIFY that only prioritized variants by zero-shot fitness prediction with no consideration of diversity, resulted in a less diverse library (Fig. 3c,d). We further compared MODIFY with DeCOIL[35], a recent ML-assisted library design method, and HotSpot Wizard[36], and observed that the MODIFY library had both higher mean experimental fitness and higher diversity (Supplementary Fig. 4; Supplementary Information A.5). Extending MODIFY to design a fifteen-site combinatorial library for the fluorescent protein CreiLOV[33] mirrored these findings (Supplementary Fig. 5; Supplementary Information A.6), underpinning MODIFY's effectiveness in striking an advantageous diversity-fitness balance across different protein families.

Furthermore, we explored MODIFY's informed setting (denoted as MODIFY-informed; Fig. 1b) and showed how prior knowledge of a protein's fitness can be incorporated through MODIFY's residue-level diversity control. In this experiment, we assumed that the experimentally determined fitness data of GB1 single mutants is available to inform the design of high-order mutants. Given the linear scaling of single-site substitutions with the number of mutation sites (20 × 4 = 80 mutants for GB1), obtaining such data is experimentally feasible and cost-effective. We observed a disparity when comparing MODIFY's zero-shot predictions to empirical fitness at position D40—MODIFY predicted all 19 possible substitutions at this site to be disadvantageous, while experiment data indicated the opposite for most mutations (Fig. 3f, g). This discrepancy showcases a potential misalignment between broad evolutionary patterns captured by zero-shot

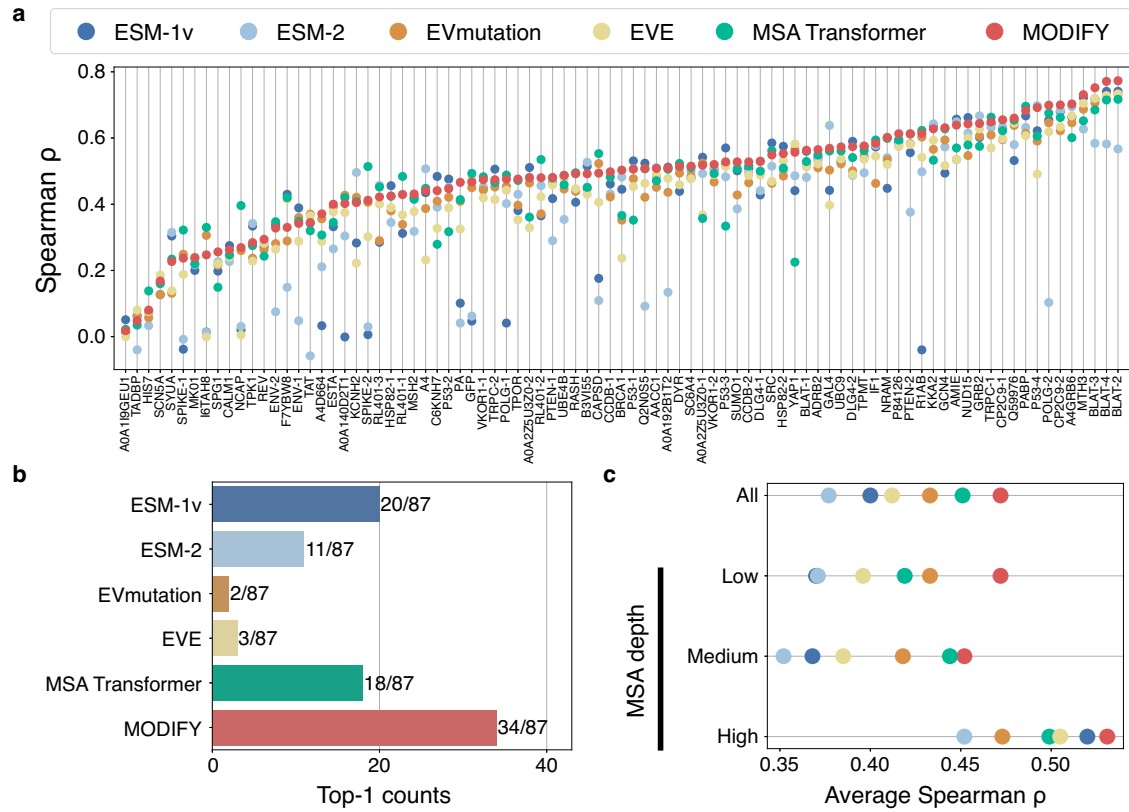

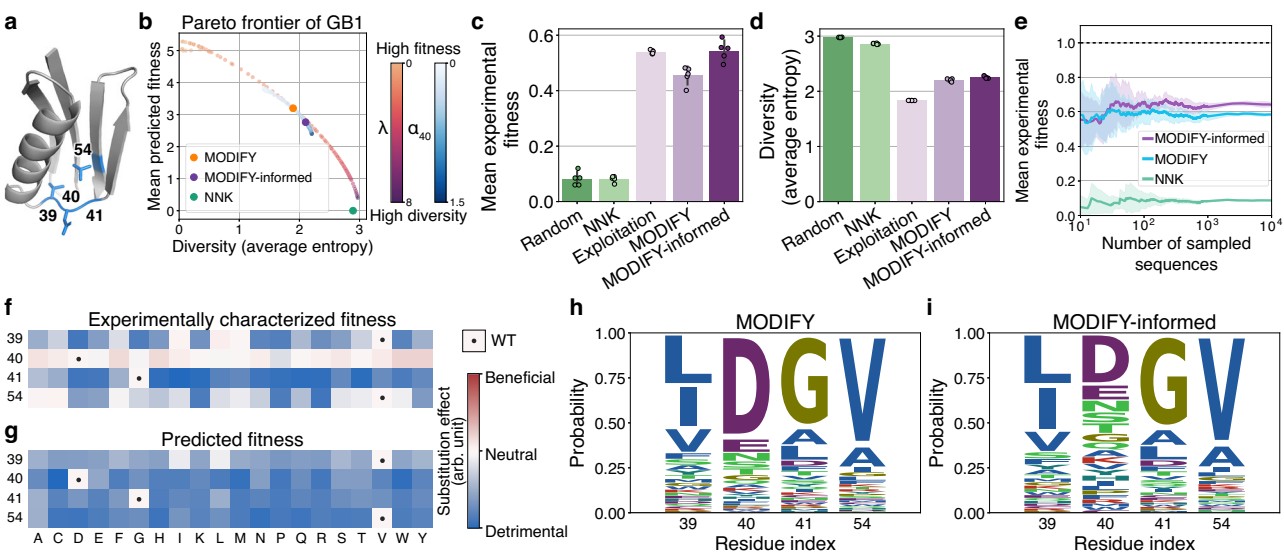

**Fig. 2 | MODIFY achieves accurate and robust zero-shot protein fitness prediction.** The ensemble ML model of MODIFY was compared with five state-of-the-art unsupervised protein fitness predictors (ESM-1v, ESM-2, EVmutation, EVE, and MSA Transformer) for zero-shot protein fitness predictions. **a** Comparison on the ProteinGym benchmark, which contains 87 Deep Mutational Scanning (DMS) assays across diverse protein families, using Spearman correlation as the evaluation metric. **b** The counts of each method achieving the best performance (including ties) on the 87 DMS datasets. **c** The average performances of all methods on proteins with low, medium, and high MSA depths.

**Fig. 3 | MODIFY designs high-quality combinatorial starting libraries for GB1.** **a** The 3D structure (PDB: 1PGA) of GB1. The four residues mutated to create combinatorial libraries are colored in blue. **b** The Pareto frontier of MODIFY library designs on GB1, with each point representing a library corresponding to a diversity strength $\lambda$. Blue points are MODIFY-informed designs by varying the residue-specific diversity weight $\alpha_{40}$ for residue 40 while fixing other weights. **c**, **d** The mean experimental fitness and diversity (average entropy) of the designed libraries, each with 500 GB1 variants. Random sampling, NNK, and Exploitation are included as the baseline methods. The bar plots represented the mean ± SD over 5 independent repetitions. **e** The mean experimental fitness of sequences sampled from the library distribution of NNK, MODIFY, and MODIFY-informed, respectively. The curves and error bands represent mean ± SD over 5 independent repetitions. **f, g** Experimentally measured fitness and MODIFY's zero-shot fitness predictions of single-mutation GB1 variants. arb. unit, arbitrary unit. **h, i** The amino acid (AA) distribution of the MODIFY and MODIFY-informed libraries.

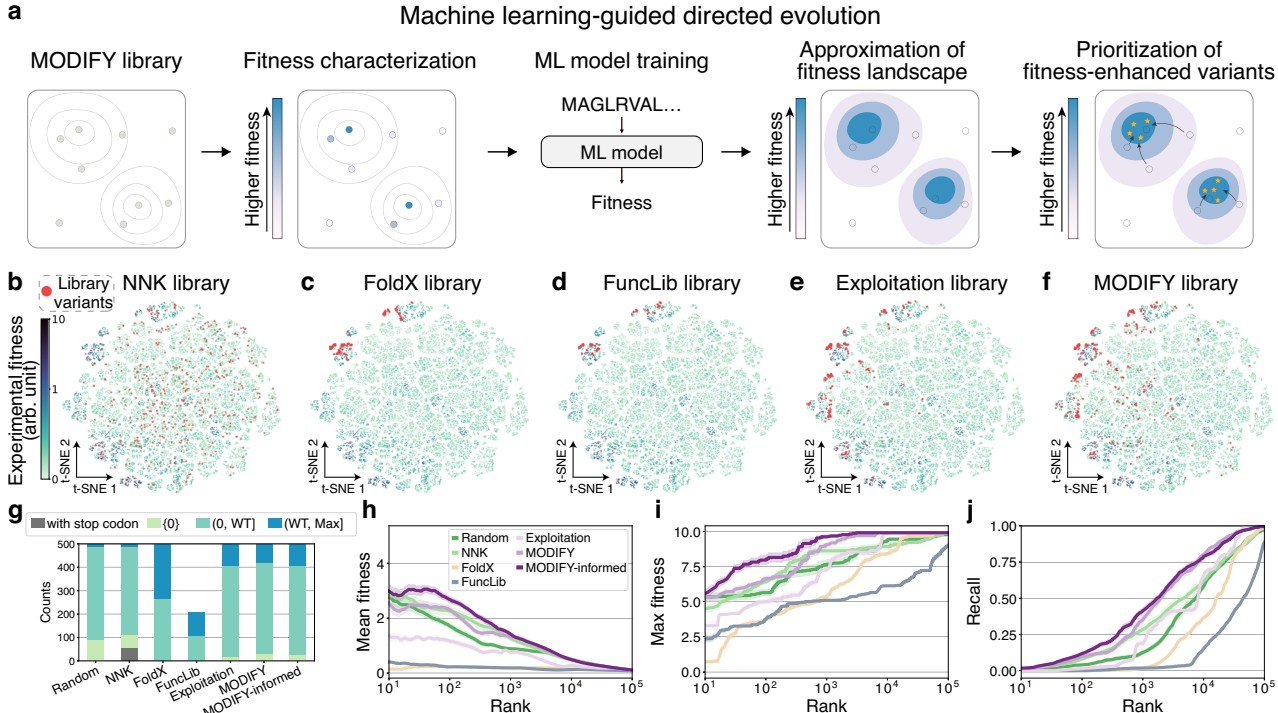

**Fig. 4 | MODIFY library improves the performances of machine learning-guided directed evolution (MLDE) on GB1. a** An in silico MLDE experiment was simulated on the GB1 landscape, where an ML model was trained to predict the sequence-fitness relationships using the variant sequences in the MODIFY library and their associated experimentally characterized fitness as the training data. The trained ML model was then applied to prioritize novel fitness-enhanced variants. **b–f** t-SNE visualization of the library sequences in the GB1 fitness landscape. Variants from various libraries (NNK, FoldX, FuncLib, Exploitation, and MODIFY) were colored in

red. arb. unit, arbitrary unit. **g** Stratified bar plots of library sequences based on their fitness ranges: (WT, Max]: better than the wildtype, (0, WT]: lower than wildtype but higher than 0, {0}: zero fitness, with stop codon: variants with stop codons. **h–j** The performance of ML models trained on fitness-labeled sequences from each library. The mean fitness (**h**), the max fitness (**i**), and the recall of the top 100 variants (**j**) as a function of the top $K$ prediction were shown. The curves and error bands represent mean ± SEM over 25 independent repetitions.

predictive models such as PLMs and the specific fitness determinants for a given protein, which resulted in the overrepresentation of the wild-type amino acid (AA) over other possibly beneficial mutations (Fig. 3h, residue 40). To counteract this effect, MODIFY-informed strategically increased the diversity weight $\alpha_{40}$ (Supplementary Information A.5), promoting the diversity of AAs for residue 40. While beneficial single mutations may not always translate into high-order mutants with improved fitness due to negative epistasis[37], promoting the diversity on site D40 may increase the chances of discovering diverse and functional four-site mutants. This approach was validated by our evaluation: the informed library (Fig. 3b) not only induced a higher diversity at both residue and sequence levels (Fig. 3d, i) but also achieved a higher mean experimental fitness compared to MODIFY (Fig. 3c, e).

Together, this study highlights MODIFY's strength in creating combinatorial libraries that effectively balance fitness with diversity. In contrast to many current library design methods[34,35], MODIFY further introduces residue-level diversity control, allowing the integration of prior knowledge into the library design process. In addition to DMS fitness data, other forms of prior knowledge, such as active-site residue effects revealed by biocatalysis data, can also be incorporated to tailor MODIFY library design.

**MODIFY library improves downstream MLDE**

The sequence composition of screening libraries plays a crucial role in MLDE, as the paired sequence-fitness data is used to train supervised ML models to guide further directed evolution experiments. To probe the impact of MODIFY library on MLDE, we simulated an in silico MLDE experiment using the GB1 landscape (Fig. 4a). We first designed a 500-

variant library using five methods: MODIFY, Exploitation, NNK, FoldX[38]—a biophysical stability prediction model, and FuncLib[39]—an automated method for designing combinatorial mutations at enzyme active sites. For FoldX, the top 500 mutants predicted to be most stable (lowest $\Delta\Delta G$) were selected. For FuncLib, all 209 designed mutants were included (Supplementary Information A.5). We then mapped the fitness landscape of GB1 onto a 2D t-distributed stochastic neighbor embedding (t-SNE) plot for an intuitive view of library composition (Supplementary Information A.5). We observed that the NNK library sampled sequences evenly scattered across the entire landscape, but most of them are low-fitness variants (Fig. 4b) with some sequences including stop codons (Fig. 4g). The FoldX library and the FuncLib library were enriched in a single high-fitness region with limited sequence diversity (Fig. 4c, d). In contrast, the MODIFY library contained variants enriched for multiple fitness peaks (Fig. 4f), suggesting a Pareto optimal library with higher mean fitness than the NNK library (Fig. 4b) and higher diversity than the FoldX, FuncLib, and Exploitation libraries (Fig. 4c–e).

Next, we paired sequences from the five libraries with their ground-truth fitness[30] and used this data to train an ML model to predict the sequence-fitness relationship. In this study, all five methods used the same encoding strategy (one-hot) and ML model architecture (random forest regressor), and the only difference was sequences in the training set defined by each library. Using these models, we predicted fitness for a set of withheld variants (Supplementary Information A.5) and ranked them accordingly. We then compared the five methods with respect to the true fitness values (mean and maximum) and the recall of the top 100 variants within their top $K$ predictions. This provided a measure of an ML model's hit rate in MLDE given a test

budget of $K$ sequences. The ML model trained on the MODIFY library outperformed all others, exhibiting the highest mean/max fitness and recall for high-fitness variants (Fig. 4h–j). Interestingly, although the FoldX library and the FuncLib library contained a higher fraction of better-than-wildtype variants (Fig. 4g), the ML model trained on the libraries performed consistently the worst with regard to mean/max fitness and recall (Fig. 4h–j). This result underscored the importance of maximizing diversity in library design for exploring the sequence space. Overall, this in silico MLDE experiment suggested that the high-fitness, high-diversity starting libraries designed by MODIFY readily translate to improved accuracy of ML models in MLDE, thus accelerating the protein engineering process.

### Experimental validation of MODIFY led to novel biocatalysts from wild-type cytochrome *c* with excellent activity and enantioselectivity

We next experimentally validated MODIFY in the design of starting functional enzyme libraries to enable valuable biocatalytic transformations that were not known in natural enzymology[20]. In particular, we sought to design functional enzyme libraries that could simultaneously promote two stereoselective new-to-nature biotransformations, including the carbon–boron (C–B) and the carbon–silicon (C–Si) bond formation reactions (Fig. 5). Although organoborane and organosilane compounds are of significant value to theranostics[40] and synthetic chemistry[41], enzymes that catalyze the formation of C–B and C–Si bonds are not known in nature. Previously, through laboratory-directed evolution via iterative site-saturation mutagenesis and screening, two variants derived from wild-type *Rhodothermus marinus* cytochrome *c* (*Rma* cyt *c*), a small thermophilic heme protein whose native function is electron transfer (Fig. 5a)[42], were separately evolved to catalyze C–B[43] and C–Si[44] formation. Each triple mutant arose from three rounds of directed evolution, with *Rma* cyt *c* V75R M100D M103T (denoted as the RDT variant, indicating its mutant type of amino acids; similar abbreviations will be used hereafter)[43] and *Rma* cyt *c* V75T M100D M103E (TDE variant)[44] being an effective borylation and silylation biocatalyst, respectively.

The MODIFY algorithm allowed us to sample high-fitness regions in the *Rma* cyt *c* sequence space not previously available from laboratory-directed evolution. In particular, we aim to engineer cytochrome *c* variants that can catalyze both C–B (Fig. 5b) and C–Si (Fig. 5c) bond formation with excellent efficiency and stereocontrol. The development of such generalist stereoselective enzymes to catalyze multiple biotransformations has remained a challenging task, as most evolved enzyme variants are reaction-specific. Guided by the crystal structure of *Rma* cyt *c* (Fig. 5a)[42,45], we constructed a MODIFY library focusing on sequence optimizations for the $\alpha$-helix residue 75 proximal to the heme cofactor and five flexible loop residues 99, 100, 101, 102, and 103 (Fig. 5d–e). In wild-type *Rma* cyt *c*, the M100 residue is bound to the Fe center to confer a hexacoordinate Fe[42]. To generate a catalytically active Fe center, we leveraged MODIFY's residue-level diversity control and eliminated M100 from our designed library (Supplementary Information A.7). We further enhanced the residue-level diversity at site 75 in light of the proximity of this residue to the heme cofactor (Fig. 5f). This MODIFY-designed *Rma* cyt *c* library contained the top 1000 variants (Supplementary Data 1). The gene fragment library was synthesized using the oligo-pool technology[46] and cloned into a pET–22b(+) vector with an N-terminal pelB sequence (see Supplementary Information A.7 for details). As a negative control, a randomized combinatorial library based on the NNK degenerate codon was also experimentally evaluated. In our experiments, 160 clones of the MODIFY library were randomly selected and screened in both the C-B and the C-Si bond-forming reactions in the form whole-cell biocatalysts or cell-free lysates (Fig. 5b–c; Supplementary Information A.7). Chiral HPLC analysis was performed to determine the yield and enantiomeric ratio (e.r.) of the organoborane and organosilane products.

Biotransformation results in Fig. 5i, j showed that MODIFY offered markedly improved results relative to the NNK control in both the C-B and the C-Si bond-forming processes. Specifically, in biocatalytic C-B bond formation, MODIFY library afforded a 2.2-fold higher averaged yield and a fourfold higher averaged enantiomeric ratio (Fig. 5i). In C-Si bond formation, MODIFY library provided a 1.9-fold higher averaged yield and a 1.3-fold higher averaged enantiomeric ratio (Fig. 5j). Importantly, an array of C-B and C-Si bond-forming biocatalysts with excellent activity and enantioselectivity emerged from this MODIFY library (Fig. 5g, h; Supplementary Tables 3 and 4). Interestingly, these best-performing borylation and silylation enzyme variants are 6 mutations away from the previously experimentally evolved RDT[43] and TDE variants[44], showcasing MODIFY's ability to identify novel functional variants not easily available by other means.

Notably, among the best-performing MODIFY borylation biocatalysts, the MGAANQ variant displayed a TTN of 2880 and an e.r. of 95:5 (Fig. 5g, entry 2) outperforming the experimentally evolved RDT variant (Fig. 5g, entry 1). In addition to the MGAANQ variant, four other MODIFY variants, including MLYPPT (Fig. 5g, entry 3), MQVANQ (entry 4), MESANQ (entry 5) and MELQNQ (entry 6), outperformed the RDT variant with respect to their total turnover numbers. Similarly, among the best-performing silylation biocatalysts, the SFLTNQ variant displayed a TTN of 3,320 and an e.r. of 98:2 (Fig. 5h, entry 2), outperforming the experimentally evolved TDE variant (Fig. 5h, entry 1). In addition to the SFLTNQ variant, another variant VQFPPQ also provided better TTN (Fig. 5h, entry 3) relative to the TDE variant.

Intriguingly, among these *Rma* cyt *c* MODIFY variants, many incorporated a proline (P) residue into the flexible loop, indicating a substantial change in loop conformation and dynamics among these MODIFY X→P variants[47]. Moreover, functional double-proline mutants with a proline at residues 99, 100, 101, and 102, including MLYPPT, MPQPNQ, VQFPPQ, KPWPNY, and SPIPAM, were uncovered from this MODIFY library. The altered loop conformation of these proline mutants with excellent catalytic activity and enantioselectivity represents a departure from the canonical structures of the experimentally evolved *Rma* cyt *c* RDT and TDE variants, further highlighting the power of MODIFY in revealing novel enzyme variants. Importantly, from this single-round screening, we identified a generalist *Rma* cyt *c* variant MLYPPT, which is highly active and enantioselective for both the borylation (2880 TTN, 94:6 e.r. (Fig. 5g, entry 3)) and the silylation (2,740 TTN, 97:3 e.r. (Fig. 5h, entry 6)) reactions, providing a rare example of promiscuous biocatalyst variant reminiscent of general small-molecule catalysts with broad utility.

The availability of a library of functional *Rma* cyt *c* variants for both C–B and C–Si bond formation also allowed us to interrogate the enzyme activity and enantioselectivity correlations between borylation and silylation reactions, a task not previously achievable due to the limited availability of functional variants. In general, *Rma* cyt *c* variants that are highly enantioselective for C–B bond formation were also found to be highly enantioselective for C–Si bond formation, as revealed by a Pearson correlation coefficient of 0.72 between the percentage of the major borane enantiomer and that of the major silane enantiomer (Fig. 5k). Similarly, variants exhibiting a higher activity in C–B bond formation are also usually more active in C–Si bond formation, despite a slightly smaller Pearson correlation coefficient (0.47) (Fig. 5l). Together, the ability to profile enzyme variant activity and selectivity in two biocatalytic reactions offer a rare opportunity to shed light on mutational effects on multiple fitness landscapes.

### MD simulations offer insights into altered loop dynamics of MODIFY *Rma* cytochrome *c* variants

To gain further insights into the flexible loop dynamics of the newly uncovered protein mutants, we carried out molecular dynamics simulations of the Fe carbene intermediates of these cytochrome *c* variants without and with NHC-BH$_3$ and PhMe$_2$SiH substrates in the

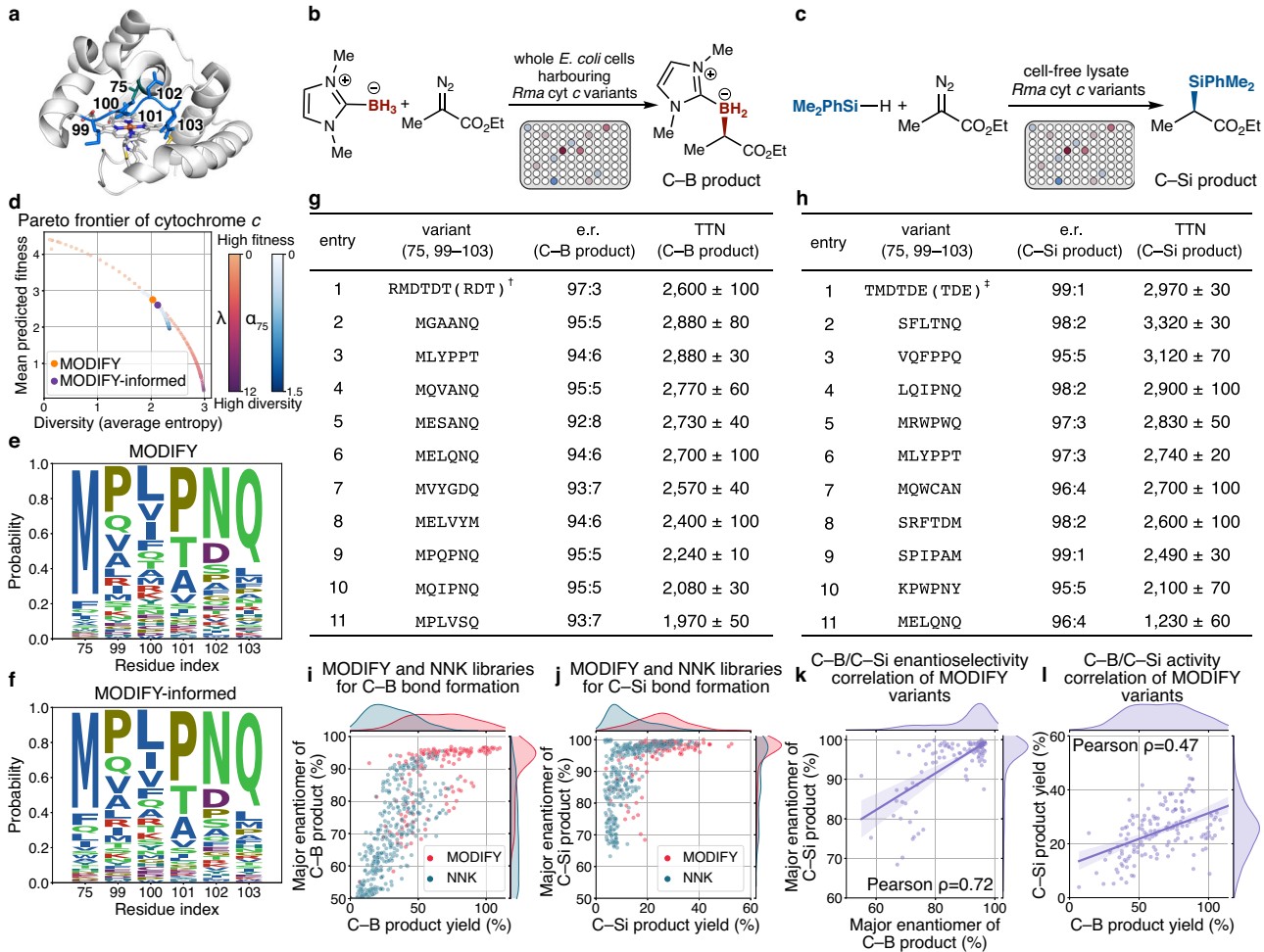

**Fig. 5 | MODIFY library of *Rma* cytochrome *c* led to new and distinct C-B and C-Si bond-forming biocatalysts. a** Crystal structure (PDB: 6CUK) of *Rma* cytochrome *c*. The six targeted residues 75, 99, 100, 101, 102, and 103 to create combinatorial libraries are colored in blue. **b** Biocatalytic carbon-boron (C-B) bond formation. **c** Biocatalytic carbon-silicon (C-Si) bond formation. **d** The Pareto frontier of MODIFY library designs on *Rma* cytochrome *c*. **e, f** The AA distribution of MODIFY and MODIFY-informed libraries. **g** Top 10 variants for C-B bond formation reactions from the MODIFY-informed library. †Previously engineered borylation biocatalyst via three rounds of directed evolution. **h** Top 10 variants for C-Si bond formation reactions from the MODIFY-informed library. ‡Previously engineered

silylation biocatalyst via three rounds of directed evolution. **i, j** Total activity (yield) and enantioselectivity (fraction of the major enantiomer) of the variants from MODIFY-informed and NNK libraries for the biocatalytic C-B and C-Si bond formation reactions, respectively. **k, l** The enantioselectivity and activity correlations between biocatalytic C-B bond formation and C-Si bond formation for the variants from the MODIFY-informed library. The error band indicates the 95% confidence interval of the regression line. TTN = total turnover number. e.r. = enantiomeric ratio. For formatting purpose, MODIFY-informed library was denoted as MODIFY in **i-l**.

active site for selected best-performing MODIFY variants and previously evolved RDT and TDE variants. Previous studies revealed the key role of this flexible loop as the dynamic lid flanking the active site of this compact heme protein and regulating catalysis[43]. Our MD simulations reveal significant changes in front loop (99-103) conformations and dynamics of MODIFY variants (Fig. 6). To quantify the flexibility of each variant, B-factor values[48] ($B_i$, Å²) were calculated from root-mean-square fluctuation ($\rho_i^{rmsf}$) of Cα atoms in MD simulations. For the Fe carbene intermediates of TDE and RDT variants (Fig. 6a), the front loops are moderately rigid, as indicated by the blue-colored region. Although the front loop dynamics of MPQPNQ remain similar to TDE and RDT variants, both the MLYPPT and SPIPAM mutants show enhanced flexibility of the front loop, as indicated by yellow to red colors. Interestingly, for the MLYPPT variant, a substantial flexibility increase is also observed for α-helix residues 91-98. The flexibility enhancement of the front loop could allow the enzyme to better accommodate the NHC-BH₃ or PhMe₂SiH substrate, leading to improved reaction efficiency. To model the substrate near-attack-conformations[49] that promote the borylation and silylation, the

distance between the carbene carbon and hydrogen atom of each substrate is restrained to be within 2.4-2.8 Å (Figs. 6c, d). Unlike RDT and TDE variants, upon the binding of either the NHC-BH₃ or the PhMe₂SiH substrates, the front loop (99-103) of MLYPPT is characterized by further enhanced flexibility, allowing for the loop to change conformation for better substrate bindings[50]. The rigidity of TDE, RDT, and MPQPNQ is due to the conserved water-mediated hydrogen bond network in the front loop[50], which does not exist in MLYPPT and SPIPAM variants (Fig. 6b). The lack of hydrogen bond networks, thus, leads to the more flexible front loop of these variants, as well as the more flexible α-helix region in MLYPPT. The substantially improved loop flexibility of the MLYPPT variant to accommodate different types of substrates may contribute to its reaction generality for both the C-B and the C-Si bond-forming processes.

## Discussion

In this work, we developed MODIFY, an ML-guided framework for cold-start library design in enzyme engineering. At the core of MODIFY is the design scheme that jointly optimizes the expected fitness and

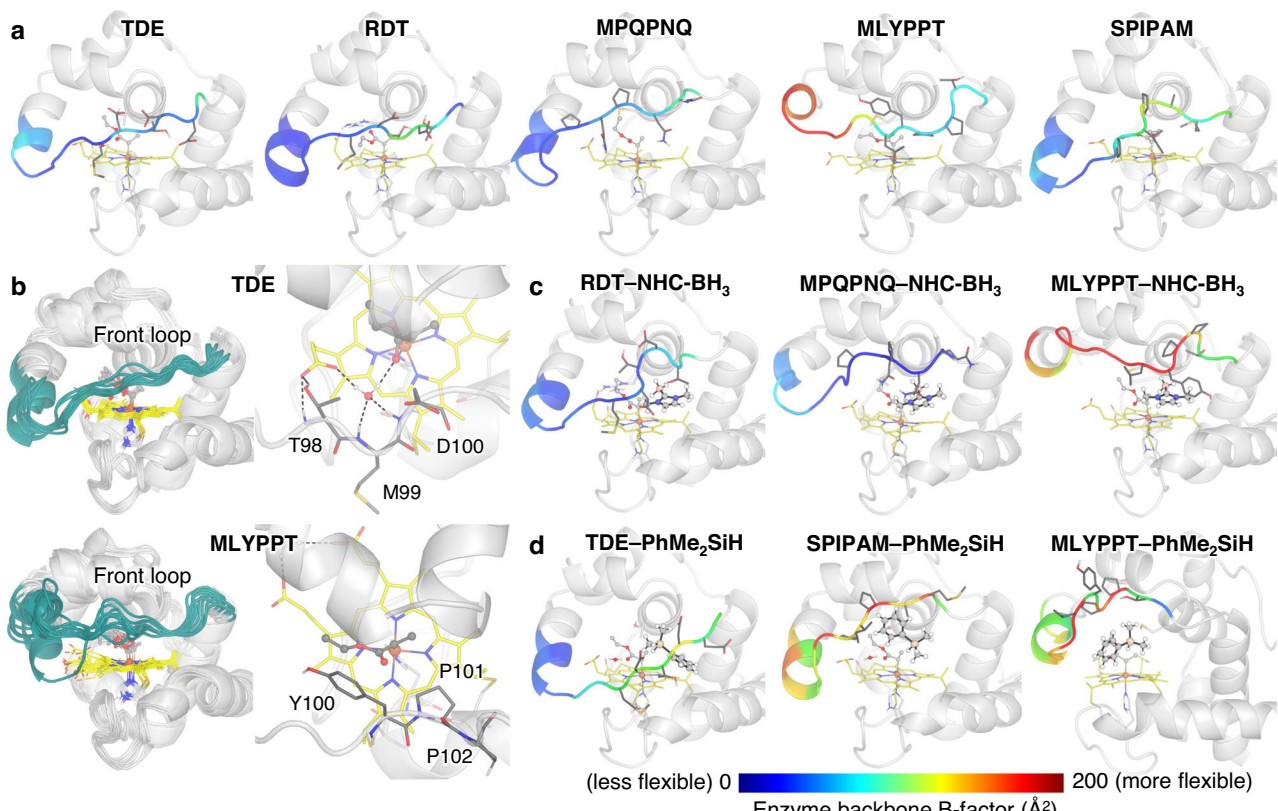

**Fig. 6 | Classical molecular dynamics (MD) simulations. a** Representative snapshots revealed contrasting loop dynamics of Fe carbene intermediates of MODIFY-designed and experimentally evolved *Rma* cyt *c* variants. **b** Overlay of 10 most-populated snapshots obtained from 1000 ns MD simulations and the presence and absence of hydrogen bond network in *Rma* cyt *c* TDE and MLYPPT variants, respectively. **c** Representative snapshots of enzyme-substrate complexes of Fe carbene with the NHC-BH₃ substrate. **d** Representative snapshots of enzyme-substrate complexes of Fe carbene with the PhMe₂SiH substrate. The backbone of residues 91–103 is colored based on B-factor value of the Cα atom of each residue.

sequence diversity of a library to achieve an optimal trade-off. To address the cold-start challenge in engineering new-to-nature enzyme functions, we developed an ensemble ML approach that integrates evolutionary information to provide zero-shot fitness prediction for protein variants. Computational evaluations suggest that MODIFY provides accurate and robust zero-shot protein fitness prediction. In silico studies using two experimentally characterized fitness landscapes showed that MODIFY provides diverse combinatorial libraries enriched with high-fitness variants and offers a more informative training set for effective MLDE.

Moreover, the MODIFY algorithm allowed the identification of novel and highly effective enzyme variants to catalyze new-to-nature reactions, permitting the enantioselective biocatalytic synthesis of useful organoboron and organosilane compounds. Top hits from the MODIFY library are six mutations away from previously engineered biocatalysts via laboratory-directed evolution. Furthermore, MODIFY led to a generalist MLYPPT variant capable of promoting both C−B and C−Si bond formation with enantiocontrol and efficiency that are superior or comparable to experimentally evolved enzyme variants. Such enzyme variants that catalyze multiple reactions with uniformly excellent activities and stereoselectivities remain rare in new-to-nature biocatalysis. MD studies revealed the diverse loop dynamic behaviors of *Rma* cyt *c* variants from the MODIFY library, indicating the excellent potential of this highly functional, highly diverse biocatalyst library in the discovery of other new-to-nature reactions. In particular, the generalist biocatalyst variant MLYPPT features unusually high degrees of loop flexibility, better accommodating both the borane and silane substrates. This finding has broad implications beyond the development of effective borylation and silylation biocatalysts.

Our work demonstrated the effectiveness of leveraging protein evolutionary data using ML for cold-start enzyme library design in new-to-nature biocatalysis. Without relying on the mechanistic considerations of newly designed enzymatic processes, the evolutionary information reveals the sequence patterns deemed by nature as prerequisites for function, such as those related to structural stability or biophysical properties. By harnessing evolutionary information, MODIFY excludes detrimental mutations and designs effective biocatalyst libraries for the development of synthetically useful enzyme functions. While experimentally assayed fitness data can be used to better guide starting library design[34], the cold-start setting considered in this work represents an essential task in diverse enzyme engineering applications, especially when large sequence-fitness datasets are not readily available.

Another critical finding of our study is the importance of sequence diversity in library design. Starting libraries with both high expected fitness and diversity are more likely to encompass variants spanning multiple fitness peaks (Fig. 4b−f), thereby enabling the navigation of regions in the fitness landscape that are challenging to reach through traditional directed evolution methods (Fig. 5). Compared to prior library design methods[15,51–54], including two studies contemporaneous to our work[34,35] that also recognized the significance of library diversity, MODIFY generates enzyme libraries without relying on known fitness data for the optimization of valuable enzyme functions, including new-to-nature functions. MODIFY further expands the optimization of sequence-level diversity[34,35,55] to residue-level diversity, allowing exploration or exploitation at specific key residues by leveraging prior biochemical and structural insights. In contrast to the previously employed degenerate-codon libraries[34,35], we implement

MODIFY libraries using microarray-based oligonucleotide pool technologies[46,56], allowing individually designed MODIFY variants to be synthesized without undesired random recombination. In enzyme engineering applications, MODIFY holds the promise to complement or replace widely implemented random mutagenesis based on degenerate codon methods.

Beyond enzyme engineering, the ML framework of MODIFY readily extends to the general field of protein engineering. Additionally, although this work focused on starting library design for single-round variants screening, MODIFY can be integrated with supervised fitness prediction models, active learning methods, and adaptive sampling algorithms to develop an ML framework for iterative-round, MLDE. This potential establishes MODIFY as a versatile tool for accelerating protein engineering, thereby enriching the already vast repository of functional proteins derived from both natural and laboratory evolution.

## Methods

### Datasets for computational evaluation

We used a diverse array of mutational scanning datasets from previous studies to computationally evaluate MODIFY's ability for zero-shot protein fitness prediction and starting library design. For zero-shot protein fitness prediction, we used the ProteinGym benchmark dataset[23] comprising 87 DMS assays with experimental measurements of protein fitness to examine MODIFY's prediction ability for single mutants. We also benchmarked on the latest release of ProteinGym (v1.0)[29]. Additionally, we included three experimentally characterized fitness landscapes (GB1[30], ParD3[31,32], CreiLOV[33]) to examine MODIFY's predictive ability for high-order mutants. For starting library design, we used the experimentally characterized fitness landscapes of GB1 and CreiLOV to examine MODIFY's ability to design high-quality combinatorial starting libraries. Detailed information on the datasets employed for computational evaluation is included in Supplementary Information A.1.

### Pareto-optimization of fitness and diversity for library design

MODIFY's primary innovation lies in its Pareto optimization algorithm for library design. For a parent protein (e.g., the wild-type or an initial functional variant) and its $M$ residues of interest, MODIFY designed a combinatorial library with optimized sequence fitness and diversity. These $M$ residues are typically chosen by protein engineers based on known functional hotspots of the protein, such as enzyme active site residues.

Denote $\mathcal{X}$ as the set of all possible sequences with length $M$ that can be included in the library, a library of protein sequences can be mathematically described using a probability distribution $p(\boldsymbol{x})$ over $\mathcal{X}$, where $\boldsymbol{x} = (x_1, ..., x_M) \in \Sigma^M$ is a sequence with length $M$ and $\Sigma$ is the set of the 20 canonical amino acids. Inspired by a recent library design method[34], we formulate the library design problem as an optimization problem of maximizing the expected fitness of sequences sampled by the library and the library's sequence diversity.

$$\max_{p \in \mathcal{P}} \mathbb{E}_{\boldsymbol{x} \sim p(\boldsymbol{x})} \text{ fitness}(\boldsymbol{x}) + \lambda \cdot \text{ diversity}(p), \qquad (1)$$

where $\mathcal{P}$ is the set of all possible libraries, and $\lambda > 0$ is a coefficient that balances the fitness and diversity terms. Therefore, the library design problem is reduced to three sub-tasks, including how to (i) estimate the fitness term fitness($\boldsymbol{x}$), (ii) quantify the library diversity diversity($p$), and (iii) efficiently find an optimal solution $p^*$ that maximizes the objective. We describe our approaches for these tasks below.

### Fitness prediction

The first challenge in high-quality library design is to efficiently predict the fitness of protein variants for guiding the inclusion of variants likely functional in the library. Although Since we

focus on cold-start library design without any known fitness data, previous approaches utilizing supervised fitness predictors are not applicable to this problem[34,57]. Here, we develop an unsupervised ML model to predict a variant's fitness $f(\boldsymbol{x})$ from its sequence $\boldsymbol{x}$. We leveraged pre-trained generative ML models, which were shown effective for zero-shot mutation effect prediction[24,27,58–60], to derive a proxy score of variant fitness (Fig. 1a). Details of this fitness predictor are provided in the next section.

### Library diversity quantification

A straightforward way to quantify the diversity of a library $p$ is through entropy $H(p) = -\sum_{\boldsymbol{x} \in \mathcal{X}} p(\boldsymbol{x}) \log p(\boldsymbol{x})$, which was adopted by a recent work[34]. However, this diversity metric, defined at the sequence level, is not sufficiently flexible to allow the incorporation of prior domain knowledge in library design; for example, some residues are critical for structure stability and should be restricted to a certain amino acid (AA), while some residues are known as tolerable for mutations so the exploration of more AAs is preferred. To address this, we delineate the sequence-level diversity into residue-level resolution. Specifically, we represent the library by a column stochastic matrix $p \in [0, 1]^{M \times K}$ such that $\sum_{k=1}^{K} p_{i,k} = 1, \forall i \in [M]$, where $K = |\Sigma| = 20$ is the alphabet size of AAs. The probability of a sequence $\boldsymbol{x} = (x_1, ..., x_M) \in \Sigma^M$ (only including AAs at the $M$ mutated sites) sampled by the library can thus be factorized as

$$p(\boldsymbol{x}) = \prod_{i=1}^{M} \sum_{k=1}^{K} \delta_k(x_i) p_{i,k}, \qquad (2)$$

where the Kronecker function $\delta_k(x_i) = 1$ if $x_i$ equals the $k$-th AA from the alphabet $\Sigma$ and zero otherwise. Accordingly, the library diversity diversity($p$) in our framework is defined as the sum of site-wise entropy of the distribution over AAs:

$$\text{diversity}(p) = \sum_{i=1}^{M} H(p_i) = \sum_{i=1}^{M} \sum_{k=1}^{K} -p_{i,k} \log p_{i,k}. \qquad (3)$$

### Optimization

Directly learning the values in matrix $p \in [0, 1]^{M \times K}$ is challenging due to the constraint that each column of $p$ should be a valid probability distribution. We thus reparameterize matrix $p \in [0, 1]^{M \times K}$ using another matrix $\phi \in \mathbb{R}^{M \times K}$ following Zhu et al.[34]: $p_{i,k} = \exp(\phi_{i,k})/\sum_{k'} \exp(\phi_{i,k'})$. In this way, we turned the constrained optimization with respect to $p$ into an unconstrained optimization with respect to $\phi$:

$$\max_{\phi} J(\phi) = \max_{\phi} \mathbb{E}_{\boldsymbol{x} \sim p(\boldsymbol{x})}[f(\boldsymbol{x})] + \lambda \sum_{i=1}^{M} \alpha_i H(p_i), \qquad (4)$$

where $\alpha_i$ is the parameter used for strengthening or reducing the diversity at residue $i$. By default, we set $\alpha_i = 1/M, \forall i$, while in the 'informed setting' (Fig. 1b), researchers can vary $\alpha_i$ to control residue-level diversity, which cannot be achieved by many existing library design algorithms[34,35,55]. This objective can be efficiently optimized by stochastic gradient ascent $\phi^{t+1} = \phi^t + \eta \nabla_\phi J(\phi)$ where $t = 1, ..., T$ is the update step and $\eta$ is the step size. Similar to Zhu et al.[34], we apply Monte Carlo approximation to estimate the gradient $\nabla_\phi J(\phi)$ using a batch of samples with size $B$:

$$\nabla_{\phi_{i,j}} J(\phi) \approx \frac{1}{B} \sum_{b=1}^{B} f(\boldsymbol{x}^{(b)})(\delta_j(x_i^{(b)}) - p_{i,j}) - \lambda \alpha_i \sum_{j'=1}^{K} (1 + \log p_{i,j'}) p_{i,j'} (\delta_j(j') - p_{i,j}), \qquad (5)$$

where $x_i^{(b)}$ is the $i$-th AA of the $b$-th sequence in the batch (Supplementary Information A.2). In our experiments, we iteratively optimized the objective for $T = 2000$ steps with step size $\eta = 0.1$ and batch size $B = 1000$.

**Pareto optimality.** The parameter $\lambda$ in Eq. (4) controls the balance between the library's expected fitness and diversity. Optimizing this objective for a specific $\lambda$ gives a particular library $p_\lambda$, corresponding to a point on the Pareto-optimal frontier curve (Fig. 1b). Points lying on this curve represent Pareto optimal solutions for the library design problem, meaning that no other solution exists that is better in terms of both mean predicted fitness and diversity. In our implementation, we used a discrete series of $\lambda$ values to recover the Pareto-optimal front, but more efficient approaches can also be employed to trace out the front[61].

**Library construction.** After deriving the Pareto front from the objective (Eq. (4)), we proceed to construct a library with $N_s$ sequences. The choice of library size $N_s$ is typically informed by experiment constraints. Since no single point on the Pareto front strictly outperforms all others, we use a heuristic approach to select a representative point $p^*$ that maximizes the area (fitness × diversity) under the curve. Next, we sample sequences from distribution $p^*$ by sequentially sampling an AA for each site from $p_i^*$ (the $i$-th column of $p^*$) and then concatenating the sampled AAs. This process repeats until the library accumulates $N_s$ sequences. In each sampling step, a sampled sequence is added to the library if it is unique to current library sequences and passes two structure filters (Fig. 1c; Supplementary Information A.4) based on foldability (ESMFold pLDDT[25]) and structure stability (FoldX $\Delta\Delta G$[38]), which improves the synthesizability of sampled sequences.

## An ensemble ML model for zero-shot protein fitness prediction

Our objective for library design (Eq. (4)) necessitates an effective method to predict the fitness $f(x)$ of a variant $x$. We harness unsupervised generative ML models in protein biology, notably protein large language models (LLMs), for zero-shot fitness prediction (i.e., no supervised training). Those LLMs, having been trained on massive protein sequence datasets, capture evolutionary patterns in natural proteins, identifying mutations that are evolutionarily plausible[58] and critical for functionality. While these patterns might not directly dictate specific functions, they reflect the mutation patterns that are prevalent across protein families or those deemed by nature as prerequisites for most protein functions; for instance, some mutations destabilize the structure and prevent the protein from carrying out a function. A recent study[58] affirmed the efficacy of LLMs in suggesting beneficial mutations for evolving human antibodies without needing function-specific data. Inspired by those findings, we utilized the evolutionary plausibility captured by unsupervised ML models as a surrogate for protein fitness, which guides our library design to sidestep fitness holes in the protein fitness landscape[62] while including those with potential high fitness. Specifically, MODIFY integrates the following four pre-trained unsupervised ML models that capture global evolutionary contexts (natural protein sequences) or local evolutionary contexts (homologous sequences). Details on how we ran those models can be found in Supplementary Information A.3.

**Protein language model.** Consider a protein sequence $x = (x_1, ..., x_L) \in \Sigma^L$, where $\Sigma$ is amino acids (AAs) set and $L$ the sequence length. A PLM predicts the probability of an AA at a specific position given the rest of the sequence: $p(x_i|x_{-i})$, where $x_{-i}$ are AAs other than $x_i$. We employed two leading PLMs, ESM-2[25] and ESM-1v[24]. Trained on the UniRef protein sequence database[63], these models capture variations across millions of observed natural protein sequences and have effective zero-shot fitness prediction ability[24,58]. To evaluate a mutant's evolutionary plausibility as a fitness proxy, we compute the log-odds ratio between the wild-type sequence $x^{WT}$ and mutant sequence $x^{MT}$ predicted by ESM:

$$s_{ESM}(x^{MT}) = \sum_{t \in T} \log p(x_t = x_t^{MT}|x_{\setminus T}) - \log p(x_t = x_t^{WT}|x_{\setminus T}), \quad (6)$$

where $T$ is the set of mutated sites and $x_{\setminus T}$ represents sequence $x$ with residues in $T$ masked. When $s_{ESM}(x^{MT}) > 0$, the mutant $x^{MT}$ is deemed more evolutionarily more plausible than the wild-type, indicating a potential higher fitness. We averaged the log-odds ratios of ESM-1v and ESM-2 in our model.

**Evolutionary coupling model.** In addition to the global evolutionary contexts captured by ESM, we incorporated evolutionary coupling (EC) models, capturing local evolutionary contexts specific to the protein of interest from its homologous sequences. Widely used for analyzing protein sequences and structures[64], those EC models employ a Potts model framework to learn the 'energy' $E(x)$ for a sequence $x$ by considering both individual site preferences and co-evolutionary patterns within the sequence: $E(x) = -\sum_i e_i(x_i) - \sum_{ij} e_{ij}(x_i, x_j)$, in which the parameters $e_i$ and $e_{ij}$ are fit on the multiple sequence alignment (MSA) of the protein's homologous sequences. The probability of the sequence is thus computed by $p(x) = \exp(-E(x))/Z$, with $Z$ a normalization constant. Here, we used an EC model known as EVmutation[26,65] and quantify the evolutionary plausibility using the log-odds ratio: $s_{EVmut}(x^{MT}) = \log p(x^{MT}) - \log p(x^{WT})$.

**Latent generative sequence model.** Another class of models we used to incorporate local evolutionary contexts is the latent generative sequence models[27,66] that learns the sequence distribution of a protein family. Those models generalize EVmutation by incorporating not just site-specific and pairwise constraints but higher-order residue interdependencies through a variational autoencoder (VAE). The probability of a sequence $x$ is defined by marginalizing out the latent variable: $p(x) = \int_z p(x|z, \theta)p(z)dz$. This is approximated using the evidence lower bound (ELBO; Supplementary Information A.3). We used EVE, a latent sequence model previously demonstrated to effectively predict mutation effects[27]. Similar to $s_{EVmut}$, the EVE-based score $s_{EVE}$ is defined as the log ratio comparing EVE-predicted probabilities of mutant to wild-type sequences.

**MSA-based PLM.** Complementing the above models, we further incorporated MSA Transformer[28], a hybrid PLM that reinforces local evolutionary patterns in the global evolutionary contexts. While similar to ESM, MSA Transformer additionally factors in the MSA of homologous sequences of the input protein to predict the amino acid probabilities for a given sequence context: $p(x_i|x_{-i}; MSA(x))$. The MSA Transformer-based score $s_{MSATrans}$ is structured similarly to $s_{ESM}$, using a sum of log-odds ratios across mutated sites.

**Ensemble fitness predictor.** The individual scores derived from the above models have been demonstrated as effective predictors for zero-shot fitness prediction or variant effect prediction[23,24,26,27,58,66]. To further enhance their accuracy and robustness, in MODIFY we integrated those models into an ensemble fitness predictor. We first applied a z-score transformation to each individual score, normalizing them to a comparable scale (zero mean and unit variance), and then averaged the normalized scores:

$$f(x) = \beta_1 \cdot \bar{s}_{ESM}(x) + \beta_2 \cdot \bar{s}_{EVmut}(x) + \beta_3 \cdot \bar{s}_{EVE}(x) + \beta_4 \cdot \bar{s}_{MSATrans}(x), \quad (7)$$

where $\bar{s}$ is the normalized score. Individual models can be weighted with different $\beta_i$ to emphasize the importance of a particular score, but we set $\beta_i = 1/4$, $\forall i$ for simplicity in this work. In our experiments (Fig. 2), we found that, while none of these existing models consistently prevails, our ensemble model is a robust predictor and outperforms others most of the time. In ablation evaluation, we tested MODIFY against every possible subset combination of its constituent models and found that including all individual models led to the best prediction performance (Supplementary Fig. 6).

### Library amplification and sequencing

**Oligo pool amplification.** A DNA oligo pool (141 bp) containing 1000 sequences designed by MODIFY was ordered from Twist Bioscience (South San Francisco, CA). The primers used in the amplifications are:

Forward primer: GTGGTCCAGTTTACATCATG

Reverse primer: GAATTGCACGTGCTTGTTCTT

The detailed methodology for amplification is provided in Supplementary Information A.7.

**Library sequencing.** The library was sequenced by the Azenta Life Sciences (Burlington, MA) with the glycerol stock of the bacteria overnight culture.

### Reporting summary

Further information on research design is available in the Nature Portfolio Reporting Summary linked to this article.

## Data availability

This study used publicly available protein fitness datasets for computational evaluation: ProteinGym benchmark dataset [https://doi.org/10.48550/arXiv.2205.13760, 10.1101/2023.12.07.570727]; the fitness datasets of high-order mutants for three proteins, including GB1 [10.7554/eLife.16965], CreiLOV [10.1021/acssynbio.2c00662], and ParD3 [10.7554/eLife.60924, 10.1038/s41559-022-01688-0]. The following PDB structures were used: 1PGA [10.2210/pdb1PGA/pdb], 1N9L [10.2210/pdb1N9L/pdb], 3CP5 [10.2210/pdb3CP5/pdb], 6CUK [10.2210/pdb6CUK/pdb], 6CUN [10.2210/pdb6CUN/pdb]. All experimental data are available in the main text and the Supplementary Information. Plasmids encoding *Rma* cytochrome *c* variants reported in this study are available for research purposes from Y.Y. under a material transfer agreement (MTA) with the University of California Santa Barbara.

## Code availability

The source code of MODIFY is available at https://github.com/luo-group/MODIFY[67] and has been deposited to Zenodo[67] at https://doi.org/10.5281/zenodo.12715542. MODIFY was built on Python 3.9, PyTorch 2.0.0, Numpy 1.23.1, Pandas 1.4.4, Matplotlib 3.5.3, Logomaker 0.8, Seaborn 0.12.2, Biotite 0.38.0, Biopython 1.76. ESM-1v (https://github.com/facebookresearch/esm), ESM-2 (https://github.com/facebookresearch/esm), EVmutation (https://github.com/debbiemarkslab/EVcouplings), EVE (https://github.com/OATML-Markslab/EVE), and MSA Transformer (https://github.com/facebookresearch/esm) were used for zero-shot protein fitness prediction. ESMFold (https://github.com/facebookresearch/esm#esmfold) and FoldX5 (https://foldxsuite.crg.eu/) were used for structured-based filtering. For MD analysis, the following software was used: Amber 20, AutoDock 4.2, Gaussian 16, and Open-source PyMOL 3.0.

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

## Acknowledgements

This research is supported by the NIH (R35GM150890 to Y.L., R35GM147387 to Y.Y. and R35GM128779 to P.L.), the Amazon Research Award (to Y.L.), and the seed grants (to Y.L. and Y.Y.) from the NSF Molecule Maker Lab Institute (grant #2019897). M.C., Y.Z., W.H., and Y.Y. acknowledge the NSF BioPACIFIC MIP (DMR-1933487) and the NSF MRSEC Program (DMR-2308708) at the University of California Santa Barbara for access to instrumentation. This work used the Delta GPU Supercomputer at NCSA of UIUC through allocation CIS230097 from the Advanced Cyberinfrastructure Coordination Ecosystem: Services & Support (ACCESS) program, which is supported by NSF grants #2138259, #2138286, #2138307, #2137603, and #2138296, and GPU resources at the Center for Research Computing of the University of Pittsburgh. The authors acknowledge the computational resources provided by Microsoft Azure through the Cloud Hub program at GaTech IDEaS and the Microsoft Accelerate Foundation Models Research (AFMR) program. We thank Prof. Kara Bren (University of Rochester) for

providing the pEC86 plasmid for cytochrome *c* maturation and Prof. Yiming Wang (University of Pittsburgh) for critical reading of this manuscript.

## Author contributions

Y.L. and Y.Y. conceived and supervised the project. K.D. developed the MODIFY model and performed computational evaluations. M.C., Y.Z., and W.H. performed molecular cloning and biocatalytic reactions. W.H. and H.W. synthesized substrates and authentic products. K.D., Y.Z., B.K.M, and W.H. performed data analysis. B.K.M. performed molecular dynamics simulations. Y.L., Y.Y., P.L., K.D., Y.Z., W.H. and B.K.M. wrote the manuscript with input from all other authors.

## Competing interests

The authors declare no competing interests.
