## [Peer Review File · Nature Communications]

Machine learning-guided co-optimization of fitness and diversity facilitates combinatorial library design in enzyme engineeringEditorial Note: This manuscript has been previously reviewed at another journal that is not operating a transparent peer review scheme. This document only contains reviewer comments and rebuttal letters for versions considered at *Nature Communications*. Mentions of the other journal have been redacted.

REVIEWERS' COMMENTS

Reviewer #1 (Remarks to the Author):

This manuscript was initially submitted to [journal name redacted], where several issues were raised. The authors made a great effort not only in addressing those issues, but also in expanding their work with additional in silico experiments, thus validating the good potential of MODIFY in guiding the design of combinatorial libraries with a well-balanced diversity and fitness for Zero-Shot predictions. Publication of the original work is therefore recommended.

Reviewer #2 (Remarks to the Author):

I carefully read through the modifications made by the authors and appreciate their clarification and changes that significantly improve the draft. One remain question I have is related to the discussion of enzyme's action to promote catalysis. The authors pointed out the importance of improved flexibility in catalysis by say:

"The flexibility enhancement of the front loop could allow the enzyme to better accommodate the NHC-BH₃ or PhMe₂SiH substrate, leading to improved reaction efficiency."

From a thermodynamic perspective, improving the flexibility of binding loop actually increases the entropic cost of substrate binding and catalysis, leading to a higher free energy barrier or worse binding free energy. This is the reason why it was generally believed that a reasonably rigid active site will promote catalysis (see DOI: 10.26434/chemrxiv-2023-9b5k7 by Fraser and discussion from Hershlag). I would like to ask the authors to provide some explanations to reconcile why in their case the additional entropy cost from enhanced loop flexibility does not increase the overall barrier, but instead, enhance the reactivity.

Of course, I can see that enhanced flexibility can make it easy for the substrate to get inside the enzyme pocket for reaction, but this effect is expected to play a trivial role if the binding is not a rate-limiting step.

Reviewer #3 (Remarks to the Author):

The authors have carefully considered and responded to all points raised in my previous report. I have also gone through their responses to the reports of the other reviewers.

I appreciate that the authors have benchmarked their MODIFY with the most recent version of ProteinGym and against TranceptEVE and GEMME. The new results suggested that the integration of TranceptEVE and GEMME could further improve the performance of MODIFY.

The authors have also compared the quality of MODIFY libraries against stronger baselines, which suggested that MODIFY trades member quality for library diversity.

Therefore, the methodological novelty of MODIFY seems to be limited to the combination of existing models (for predicting mutational effects) and the introducing of a diversity metric for library design, both adjustments carried out in somewhat subjectively chosen ways.

The GB1 and the cytochrome C examples unfortunately did not demonstrate that MODIFY could design libraries beyond a small number of targeted residues. To design libraries for such small numbers of changeable residues may not be considered as challenging for conventional structure-based methods (if molecular docking were to be used for selecting the targeted residues, structure information should be available).

Our responses to Reviewers 1–3 are detailed below:

In response to Reviewer #1:

1. “This manuscript was initially submitted to [journal name redacted], where several issues were raised. The authors made a great effort not only in addressing those issues, but also in expanding their work with additional in silico experiments, thus validating the good potential of MODIFY in guiding the design of combinatorial libraries with a well-balanced diversity and fitness for Zero-Shot predictions. Publication of the original work is therefore recommended.”

We appreciate the suggestions of this reviewer in the revision of our manuscript. These suggestions helped improve the quality of our work.

In response to Reviewer #2:

1. “I carefully read through the modifications made by the authors and appreciate their clarification and changes that significantly improve the draft.”

We thank this reviewer for the input during the review process.

2. “One remain question I have is related to the discussion of enzyme's action to promote catalysis. The authors pointed out the importance of improved flexibility in catalysis by say:

"The flexibility enhancement of the front loop could allow the enzyme to better accommodate the NHC-BH3 or PhMe2SiH substrate, leading to improved reaction efficiency."

From a thermodynamic perspective, improving the flexibility of binding loop actually increases the entropic cost of substrate binding and catalysis, leading to a higher free energy barrier or worse binding free energy. This is the reason why it was generally believed that a reasonably rigid active site will promote catalysis (see DOI: 10.26434/chemrxiv-2023-9b5k7 by Fraser and discussion from

Hershlag). I would like to ask the authors to provide some explanations to reconcile why in their case the additional entropy cost from enhanced loop flexibility does not increase the overall barrier, but instead, enhance the reactivity.

Of course, I can see that enhanced flexibility can make it easy for the substrate to get inside the enzyme pocket for reaction, but this effect is expected to play a trivial role if the binding is not a rate-limiting step.”

Our discussion in this context is mainly to rationalize the improved generality of this MYLPPT variant. The improved loop flexibility allows the accommodation of different types of X-H substrates (NHC-BH₃ and PhMe₂SiH), leading to a more “generalist” enzyme catalysts. Additionally, in enzyme catalysis, conformational flexibility can often promote catalytic activity. This is because during catalysis, the substrate/reactive intermediate often undergoes substantial conformational changes and a flexible protein scaffold can often better accommodate these conformational changes during catalysis.

In response to Reviewer #3:

1. “The authors have carefully considered and responded to all points raised in my previous report. I have also gone through their responses to the reports of the other reviewers. I appreciate that the authors have benchmarked their MODIFY with the most recent version of ProteinGym and against TranceptEVE and GEMME. The new results suggested that the integration of TranceptEVE and GEMME could further improve the performance of MODIFY.

The authors have also compared the quality of MODIFY libraries against stronger baselines, which suggested that MODIFY trades member quality for library diversity.

Therefore, the methodological novelty of MODIFY seems to be limited to the combination of existing models (for predicting mutational effects) and the introducing of a diversity metric for library design, both adjustments carried out in somewhat subjectively chosen ways.”

In our previous response, we detailed the key innovation of our computational methods. Our ensemble method provided enhanced zero-shot prediction of protein fitness. The Pareto optimization framework allowed the generation of high-quality libraries with balanced fitness and diversity.

2. “The GB1 and the cytochrome C examples unfortunately did not demonstrate that MODIFY could design libraries beyond a small number of targeted residues. To design libraries for such small numbers of changeable residues may not be considered as challenging for conventional structure-based methods (if molecular docking were to be used for selecting the targeted residues, structure information should be available).”

The focus of this manuscript is to design fitness-diversity balanced libraries given a few selected residues. The selection of key residues for enzyme engineering is beyond the scope of this study and is ongoing in our laboratory.